# Active Learning for Natural Language Generation

**Yotam Perlitz**[*], **Ariel Gera**[*], **Michal Shmueli-Scheuer,**
**Dafna Sheinwald, Noam Slonim, Liat Ein-Dor**
IBM Research AI
{yotam.perlitz,ariel.gera1}@ibm.com,
{shmueli,dafna,noams,liate}@il.ibm.com

## Abstract

The field of Natural Language Generation
(NLG) suffers from a severe shortage of labeled
data due to the extremely expensive and time-
consuming process involved in manual annota-
tion. A natural approach for coping with this
problem is active learning (AL), a well-known
machine learning technique for improving an-
notation efficiency by selectively choosing the
most informative examples to label. However,
while AL has been well-researched in the con-
text of text classification, its application to NLG
remains largely unexplored. In this paper, we
present a first systematic study of active learn-
ing for NLG, considering a diverse set of tasks
and multiple leading selection strategies, and
harnessing a strong instruction-tuned model.
Our results indicate that the performance of
existing AL strategies is inconsistent, surpass-
ing the baseline of random example selection
in some cases but not in others. We highlight
some notable differences between the classifi-
cation and generation scenarios, and analyze
the selection behaviors of existing AL strate-
gies. Our findings motivate exploring novel ap-
proaches for applying AL to generation tasks.

## 1 Introduction

Active learning (AL) (Cohn et al., 1996) is a well-
known machine learning approach for reducing
annotation effort, aiming to train models with less
data by selecting the most informative examples
to label. This paradigm was introduced and devel-
oped in the context of classification (Settles, 2009;
Lewis and Gale, 1994), and has been successfully
applied to machine learning problems from a wide
range of domains, including computer vision (Gal
and Ghahramani, 2016; Sener and Savarese, 2018;
Gissin and Shalev-Shwartz, 2019) and text classi-
fication (Zhang et al., 2017; Siddhant and Lipton,
2018; Prabhu et al., 2019; Ein-Dor et al., 2020).

---

[*]These authors contributed equally to this work.

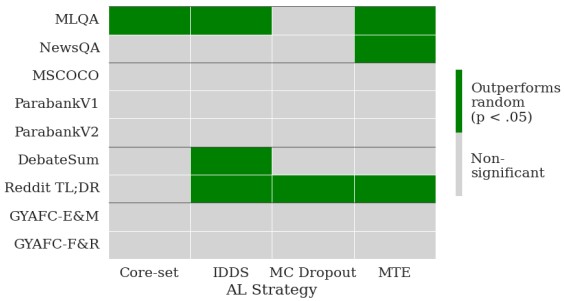

Figure 1: **Statistical significance of AL benefits.** The
plot depicts the results of paired Wilcoxon signed-rank
tests of AL performance, for each dataset-strategy com-
bination. Cells marked in green indicate that the AL
strategy significantly outperformed random selection for
the dataset, with $p < .05$ after Bonferroni correction.

Major advances in the architecture and scale
of machine learning models in general, and pre-
trained language models in particular, have given
rise to the emerging field of Natural Language Gen-
eration (NLG) (Li et al., 2021; Dong et al., 2022).
However, a major practical barrier in tackling NLG
tasks is the shortage of annotated data, exacerbated
by the increased burden of human annotation for
such tasks. As a paradigm for minimizing label-
ing effort, AL is a natural avenue for coping with
these challenges. Nevertheless, it has hardly been
studied in the context of NLG problems.

In this work, we explore the AL paradigm as
applied to NLG. Our aim is twofold: first, to exam-
ine how well AL strategies perform in NLG; and
second, to gain insight into how these strategies
operate in the unique context of text generation.
To this end, we conduct an extensive set of experi-
ments with leading AL strategies on various NLG
tasks, accompanied by a comprehensive analysis
of strategy behaviour. Our results reveal that these
AL strategies do not perform consistently across
different datasets and tasks (Figure 1), suggesting
that new AL methods are required to bring value in

this domain.

To the best of our knowledge, this is the first systematic study of AL for NLG tasks. Moreover, in this work we introduce strong instruction-tuned models into the AL paradigm. In outlining the behavior of existing strategies, we aim to lay the groundwork for future research, leading to effective use of AL in practical NLG scenarios.

## 2 Related Work

In the field of natural language processing, AL was mainly studied for text classification (Ein-Dor et al., 2020; Schröder et al., 2021; Margatina et al., 2021).

AL has also been successfully applied in the context of neural machine translation (NMT), where the focus is on low-resource pairs of languages (Zhao et al., 2020; Liu and Yu, 2023). Some works investigated strategies that are tailored specifically for NMT, such as using a backward translator to check round-trip translations (Haffari et al., 2009; Zeng et al., 2019) or quality estimation (Chimoto and Bassett, 2022). Zeng et al. (2019) conducted a systematic comparison of different AL methods in the context of NMT. Thus, we do not include NMT in the present work and focus on NLG tasks that had not been systematically explored.

There exists a wide variety of NLG tasks, and these have attracted much attention in recent years (Dong et al., 2022). Nevertheless, outside the context of NMT there are very few works that apply AL to NLG tasks (Zhang et al., 2022). Specifically, for summarization, Gidiotis and Tsoumakas (2021, 2022) propose a Bayesian strategy in which they select samples to label by optimizing the uncertainty using the Monte Carlo BLEU variance metric. More recently, Tsvigun et al. (2022) propose an embedding-based method and show improvements in certain summarization datasets. Paraphrase generation with LSTMs was reported in Karaoguz (2018), where the authors use $n$-gram coverage measures as their sampling strategy, aiming at capturing the informativeness of source paraphrases.

Thus, while there have been some focused studies of AL in the context of a specific NLG task, in this work we aim for a more systematic and comprehensive view, across multiple tasks and datasets.

## 3 Background

### 3.1 The Active Learning Scenario

Active learning is an iterative process that aims to reduce labeling costs, by focusing the human annotation effort on the most informative instances.

The AL setting assumes access to a large amount of unlabeled data, and a limited annotation budget. The core idea of AL is that the current model can be leveraged to maximize the utility of the labeling budget; thus, the goal of an AL strategy is to identify unlabeled examples whose annotation is expected to yield the largest performance improvement when used to train the model.

Differing approaches have been put forth for predicting – given a set of *unlabeled* examples – which of those examples would be most beneficial as training examples for the model.

Broadly, much of the AL literature has focused on two general directions: Representativeness and Informativeness. **Representativeness** methods focus on the data distribution of the examples. Assuming that an informative set of training examples is one that accurately represents the overall population of instances, they aim to select a diverse and representative set of examples for labeling. Under the **Informativeness** umbrella, a key focus has been on *uncertainty*. In the uncertainty-based approach, the core assumption is that examples for which the model is least certain are the most informative for model training. Thus, uncertainty-based strategies aim to estimate the uncertainty for each unlabeled example $u$, and to choose those with the highest model uncertainty.

### 3.2 AL in Generation vs. Classification

Text classification and text generation differ in many important aspects. Next, we consider these differences through the lens of AL strategies.

One major difference is that, for most NLG tasks, there are multiple legitimate outputs for a given input text. For example, in paraphrase generation, there are many ways to rephrase a given sentence; the ability to generate a diverse set of outputs is in fact a desired attribute of the model.

Generally, in AL, a model's uncertainty about an example is considered a strong signal for informativeness; the underlying assumption is that examples where the model is uncertain are those where it is prone to error, and would thus derive the most benefit from supervision. However, uncertainty in an NLG scenario – namely, a situation where the model considers several outputs to be equally probable – is not necessarily associated with errors, and may even reflect a desirable propensity to generate diverse generation outputs. Therefore, the family of

uncertainty-based active learning strategies, considered a top strategy for classification (Schröder et al., 2021), may not be directly applicable to NLG.

Another fundamental difference between classification and generation is in the dimensionality of the prediction space. In classification tasks, the number of classes is typically small, ranging from two classes, e.g., in the case of spam detection, up to a few hundreds for intent detection tasks. In contrast, in NLG, the number of possible "classes" in predicting the next token is the vocabulary size – which is typically $O(10^4)$ – and correspondingly the number of options to select from when generating a *sequence* of tokens is exponentially large. The dimension of the prediction space is crucial in the case of expected model change strategies, which aim to directly estimate the effect an instance would have on the model. While in classification a strategy like Expected Gradient Length (Huang et al., 2016) can compute the expected gradient norms over the posterior distribution of labels, the very large dimension of the prediction space in generation tasks makes this computation intractable.

# 4 Experimental Setup

This work aims to systematically study the application of active learning to NLG. To this end, we conduct a comprehensive set of experiments, exploring the behavior of the different families of AL strategies across a wide range of NLG tasks.

## 4.1 Active Learning Setup

We use the pool-based active learning (Settles, 2009) variant, in batch mode.

At the beginning of the active learning process for a dataset $D$, we start with a pre-trained base model $M_0$, a pool of unlabeled data $U_D$ and an empty pool of labeled data $L_D$.

At each active learning step $i$, the AL strategy selects a batch of $n_i$ samples from $U_D$ for labeling; these samples are removed from $U_D$ and added to the labeled data pool $L_D$, along with their ground-truth labels. Then, the base model $M_0$ is fine-tuned on the labeled samples in $L_D$, i.e., on all the data labeled thus far, creating a new model $M_i$.

This process is repeated $N$ times, where at each step the AL strategy can utilize the predictions over the unlabeled pool of the previous model $M_{i-1}$, in order to select the next batch of examples.

For runtime considerations, we restrict the size of the unlabeled pool $U_D$ to 10K examples randomly sampled from the training set of $D$.

Altogether, we report results of 18 AL steps between 0 and 1000 labeled examples: 10 batches of 20, followed by 8 batches of 100. As our focus is on a practical scenario of small annotation budgets, we opted to sample the first iterations (i.e., $0 - 200$ training examples) more densely, to gain a detailed view of the behavior of AL in this area.

## 4.2 Base Model

To represent a practical scenario of applying AL over strong pretrained models, we use the instruction-tuned FLAN-T5 Large[1] as the base model for our experiments. This model was trained on a wide range of language tasks, including many NLG tasks, and has been shown to exhibit better performance and faster convergence in downstream task fine-tuning (Longpre et al., 2023). As this base model was trained using instruction-tuning (Wei et al., 2022), we formulate each NLG task as a simple natural language instruction that is given to the model. The instruction prompts for each task are listed in Appendix A.2.

To ensure an appropriate simulation of the AL process, we only experiment on datasets that were not included in the FLAN-T5 training data[2].

Note that the use of a strong base model with zero-shot capabilities allows for starting the AL process from an empty labeled data pool $L_D$. This is unlike the traditional AL setup, where a randomly-selected seed $L_D$ is required to jump-start the process.

## 4.3 Tasks and Datasets

We consider four prominent tasks in NLG: paraphrase generation, style transfer (formality), summarization, and question generation. We chose 2 or 3 representative datasets for each task. As mentioned above, we avoid datasets that were used to fine-tune FLAN.

The datasets for each task are listed in Table 1, and a full description of the datasets can be found in Appendix A.1.

## 4.4 Active Learning Strategies

We test a representative group of AL strategies, covering different data acquisition approaches. Following Zhang et al. (2022), we divide AL query

---

[1] https://huggingface.co/google/flan-t5-large
[2] https://github.com/google-research/FLAN/blob/main/flan/v2/flan_collection_info.csv

| Task | Datasets | Metric |
|------|----------|--------|
| *Paraphrase Generation* | MSCOCO, Parabank v1.0/v2.0 | iBLEU |
| *Summarization* | DebateSUM, Reddit TL;DR | ROUGE-L |
| *Question Gen.* | NewsQA, MLQA | BLEU |
| *Formality* | GYAFC-E&M, GYAFC-F&R | G-Score |

Table 1: Datasets and evaluation metrics.

strategies into two broad categories - **representativeness** and **informativeness** strategies.

Where necessary, we adapt the strategy implementation to accommodate the NLG scenario. Note that certain types of AL strategies are inherently unsuitable for NLG. Specifically, *gradient-based* AL methods like EGL (Huang et al., 2016) or BADGE (Ash et al., 2020) cannot be straightforwardly applied to NLG, due to the sequential nature of NLG.

### 4.4.1 Representativeness Strategies

For batch-mode AL, the AL variant we use in this study, the relevant representativeness strategies are those that aim to optimize the diversity and representativeness of the selected batch. We take greedy **Core-Set** and **IDDS** as examples of this family.

**Core-Set** (Sener and Savarese, 2018) aims to increase representativeness by selecting instances with maximal distance from the labeled pool. We follow the greedy method described in Sener and Savarese (2018). As in our scenario we start from an empty labeled data pool, for the first AL step we modify this strategy to enable it to be applied without labeled data. For details see Appendix A.5.

**In-Domain Diversity Sampling (IDDS)** aims to select diverse instances while avoiding outliers (Tsvigun et al., 2022). IDDS scores for an example strike a balance between a large mean distance of the example from the instances in the labeled pool, and a small mean distance from those in the unlabeled pool.

The above strategies rely on vector representations for each instance; following Ni et al. (2022), we calculate these representations as the hidden state of the last layer of the encoder, averaged across input tokens.

### 4.4.2 Informativeness Strategies

Informativeness strategies rank unlabeled examples according to measures that estimate example informativeness, where model uncertainty is often a proxy of informativeness. We take **MTE** as an example of an *uncertainty sampling* strategy, and **MC Dropout** which is a *disagreement-based* strategy.

**Mean Token Entropy (MTE)** selects instances the model is least certain about, according to the max-entropy decision rule. For NLG, the notion of max-entropy is expanded by taking the mean over the entropies of each generated token, as in Zhao et al. (2020).

**Monte Carlo Dropout (MC Dropout)** selects instances the model is least certain about, by harnessing model stochasticity (Gal and Ghahramani, 2016). For NLG, instance uncertainty is estimated using Monte Carlo BLEU variance (Gidiotis and Tsoumakas, 2021; Xiao et al., 2020). In this approach, after stochasticity is induced by activating dropout, the uncertainty of a specific sample is estimated by how different its generated outputs are in terms of their BLEU score.

### 4.5 Evaluation Metrics

We use standard automatic metrics to evaluate the tasks, as summarized in Table 1. For paraphrase generation we use **iBLEU** (Sun and Zhou, 2012); for summarization and question generation, we use **ROUGE-L** (Lin and Och, 2004), and **BLEU** (Papineni et al., 2002), respectively. To evaluate the formality transfer task, we follow Xu et al. (2018) and use **G-Score**, the geometric mean of the formality-score and BERTScore (Zhang et al., 2020) with the reference text; further details can be found in Appendix A.6.

### 4.6 Implementation Details

We base our training and inference implementation on Hugging Face Transformers (Wolf et al., 2019) v4.26 and pytorch (Paszke et al., 2019) v2.0. Each experiment was repeated 5 times, with each repetition using a different random seed on a single NVIDIA A100 GPU. Thus, we performed a total of 4050 training and inference runs (9 datasets × 5 strategies × 18 iterations × 5 repetitions) for the main experimental results.

To keep the computation budget manageable, we opted for a single base model and standard set of hyperparameter values. In each AL step, the base model was fine-tuned for 3 epochs over $L_D$, using the adafactor optimizer (Shazeer and Stern, 2018) with a constant learning rate of $5 \times 10^{-5}$, and train batch size of 8.

## 5 Results

For each dataset, we report performance metrics across 18 AL iterations – i.e., the performance of

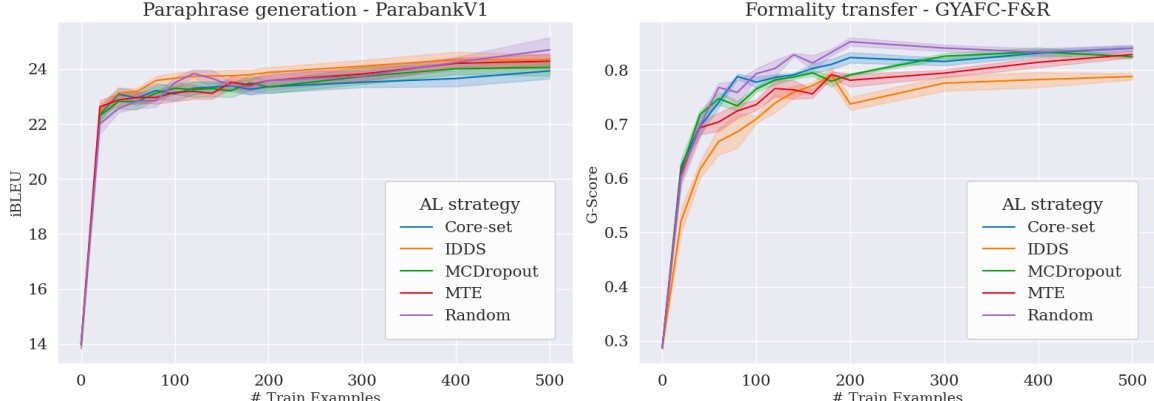

Figure 2: **AL performance for selected datasets.** The lines depict the evaluation performance of the different selection strategies along the AL process. Each line represents an average ($\pm$ 95% Bootstrapped CI) over 5 experimental repetitions. In this plot we focus on the range of up to 500 labeled examples. Plots for all datasets, for the full range of 1000 examples, are shown in Appendix A.3.

models fine-tuned on different amounts of labeled instances – comparing results between different active learning strategies.

In addition, we report the performance when using the baseline of *Random selection* - randomly sampling the batch of examples to be added to the labeled data pool at each iteration.

Figure 2 depicts AL results for two of the datasets tested, Parabank1.0 and GYAFC-F&R. As can be seen in the figure, for these datasets we do not see a clear advantage to any of the AL strategies tested; moreover, the various AL strategies do not seem to outperform the baseline of randomly selecting instances for labeling. These plots are representative of the overall pattern of results we see across tasks and datasets, with none of the AL strategies convincingly overtaking the others across datasets. Individual plots for all the datasets tested are shown in Appendix A.3.

A recurring pattern in the results is that most of the performance gains from supervision occur in the first few AL iterations, and at times even in the first AL iteration, where the model is trained on just 20 labeled instances. Thus, it appears the FLAN base model needs only a small number of examples to learn the basic premise of the target generation task; while larger amounts of labeled data are of course beneficial, the few-shot performance of this model across datasets is quite strong.

In Figure 3 we present the full distribution of the performance of the various AL strategies relative to the random selection baseline, for each of the 4 NLG tasks. Overall, across tasks, datasets, AL iterations and experiment repetitions, the behavior of

all the strategies tested is strikingly similar to that of random selection. Granted, there are specific instances where a strategy outperforms random selection for a specific dataset (for example, see the MC Dropout strategy over the Reddit TL;DR data); however, we find no consistent pattern of benefits to using AL, even within a specific NLG task.

To better quantify these results, we perform a Wilcoxon signed-rank test, comparing each strategy to the random selection baseline (refer to Appendix A.4 for details). Figure 1 shows the results of the significance testing. As can be seen, none of the strategies exhibit a clear benefit for the tasks of paraphrase generation and formality transfer, and results for question generation and summarization are somewhat mixed. Notably, both Core-Set and MC Dropout fail to provide benefits across more than one dataset.

## 6 Analysis

Given the failure to systematically outperform the random baseline, in this section we examine the different AL strategies by the properties of the batches they select, aiming to gain insights into how they operate. In line with the two families of strategies we survey, our analyses focus on notions of representativeness (§6.1) and uncertainty (§6.2).

We perform a comparative analysis of the strategies, using the random strategy as a reference point. This analysis also serves as a sanity check, to validate that the properties exhibited by the different strategies are aligned with their expected behavior.

To ensure all strategies are compared with the

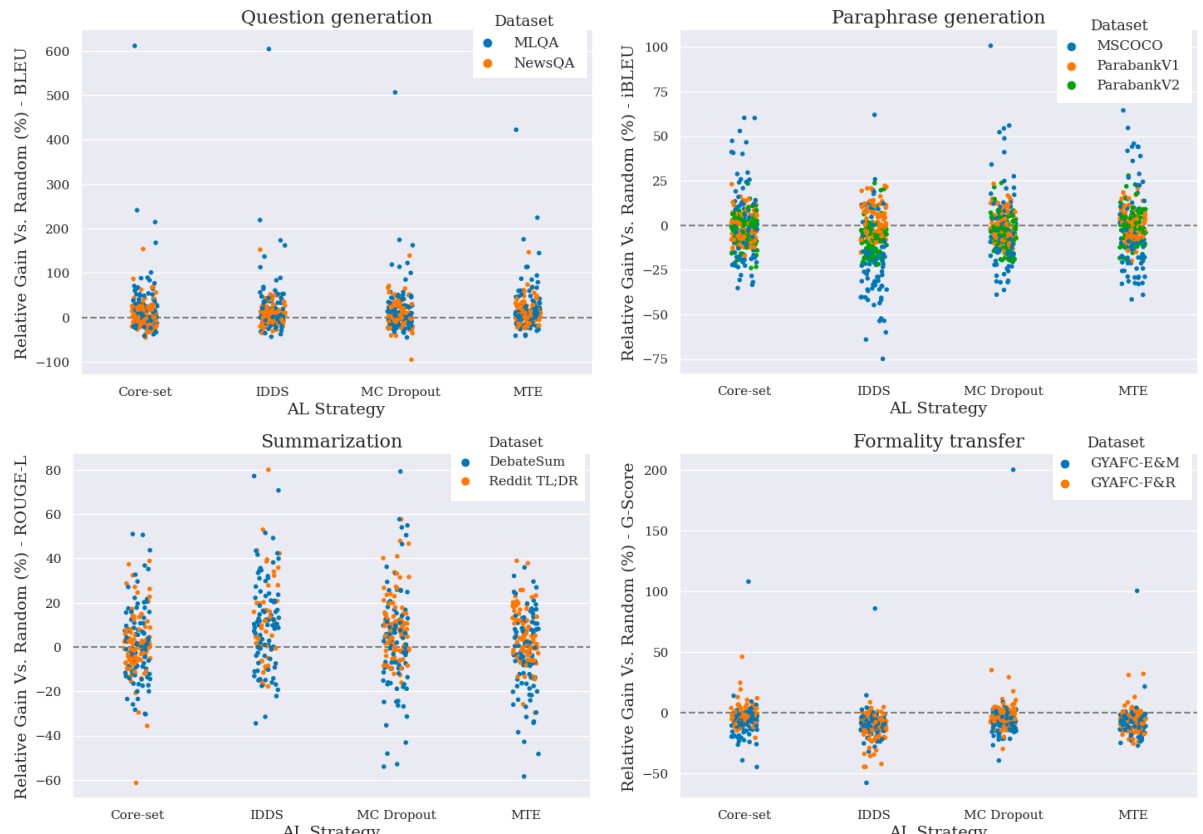

Figure 3: **Dataset/Strategy Summary.** The plots depict the *relative gains* of each strategy with respect to the random selection baseline. *Gains* are computed as the performance difference with respect to the zero-shot performance. *Relative gains* are computed as the percentage change between the *gains* of the AL strategy and the *gains* of random selection. Each point represents the *relative gain* between a given strategy and random selection for a particular setting – i.e., at a specific iteration and for a specific experimental repetition. Thus, for each dataset-strategy combination 90 points are shown (18 AL iterations $\times$ 5 repetitions). The distribution patterns reveal that although in some cases, a strategy might beat random selection, no strategy consistently outperforms the random baseline.

same initial conditions, this analysis is performed solely for the first AL iteration, where all strategies rely on the same base model and the same unlabeled set $U_D$. Each strategy makes its own selection of 100 examples[3] for labeling from $U_D$.

## 6.1 Diversity and Outliers

Two properties that are known in the literature to impact the effectiveness of AL strategies, at least in the context of classification, are the diversity and the propensity for outliers of the batches selected for annotation (Kee et al., 2018). Thus, we examine these two properties in the selected batches.

For the purpose of analyzing these properties, we define the input example representation as the average of the input tokens' embeddings over the last encoder hidden state (Ni et al., 2022), as done

for the representativeness strategies in §4.4.1.

**Outliers:** A known issue with AL strategies, particularly those focusing on uncertainty, is their tendency to select outliers that do not faithfully represent the overall data distribution. To measure the severity of the outlier problem, we use the density in representation space of points in the selected batches. Specifically, following Ein-Dor et al. (2020), we rely on the KNN-density measure proposed by Zhu et al. (2008), where the density of an instance is quantified by the average (Euclidean) distance between the instance in question and its $K$ nearest neighbors within $U_D$. We define the outlier-score of a batch by the average KNN-density of its instances ($K = 10$), where high density corresponds to a low outlier-score.

**Diversity:** Choosing a batch of diverse examples is often better than choosing one containing very similar and perhaps redundant examples. We define

---

[3]To obtain a more accurate estimate of strategy behavior, we use a larger initial batch size than for the results in §5.

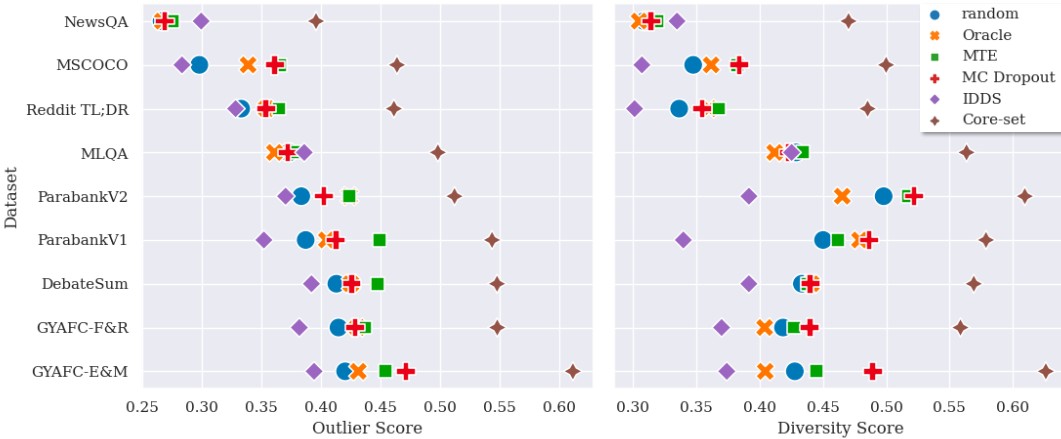

Figure 4: **Strategy selection characteristics.** The plots depict the outlier score (left, lower is better) and diversity (right, higher is better) of the batches selected by the different AL strategies, as well as the *Oracle* strategy of §6.2.2.

the Diversity of a batch $B$ as the average Euclidean distance of its instances from the center.

The diversity and outlier-score of the different strategies are depicted in Figure 4. As expected, Core-Set, a batch-aware strategy, which was designed to increase diversity, is characterized by batches with the highest diversity. It is also characterized by the highest outlier score, in line with the tendency of the greedy version of Core-Set to select outliers (Sener and Savarese, 2018). In contrast, IDDS, which explicitly attempts to avoid outliers (Tsvigun et al., 2022), correspondingly has a low outlier-score, and also a relatively low diversity. Meanwhile, the uncertainty strategies exhibit diversity and outlier scores that are closer to those of the random selection baseline, indicating that they do not suffer from severe diversity or outlier issues.

## 6.2 Uncertainty and Model Performance

The major premise underlying uncertainty-based AL is that the information gained by labeling an example is higher if the current model's prediction on this example is erroneous. Thus, relying on the potential association between uncertainty and error rate, uncertainty-based strategies aim to find examples from the unlabeled data that are more prone to errors (Lewis and Gale, 1994).

The failure of the uncertainty-based strategies examined here to consistently outperform the random baseline, raises the question if – and to what extent – they are applicable to the generation scenario.

In order for the uncertainty approach to succeed, two conditions must be fulfilled: First, that the strategies are able to identify subsets of the unla-

beled examples with a larger error rate; and second, that labeling examples with larger model errors is in fact more beneficial for training the model. The following analyses examine these two conditions.

### 6.2.1 Selecting Error-prone Examples

In classification, there exists a clear association between uncertainty and prediction errors. Here we examine whether a similar association also holds in generation; and specifically, whether examples that are scored higher by the uncertainty strategies are associated with poor generated outputs, as measured by the automatic evaluation metrics.

To this end, we obtain the generation predictions of the base model for all the instances in the unlabeled pool $U_D$. Then, we compute the average evaluation score of the model generations that correspond to instances in the selected batch $B$, and compare them to the average score over the entire unlabeled pool. The results of this analysis for the various AL strategies are presented in Figure 5. As expected, batches selected by the MC Dropout uncertainty strategy are characterized by lower evaluation scores. The other uncertainty approach, MTE, exhibits a similar tendency but is less consistent across datasets.

Thus, there is some association between generation performance and uncertainty. Nevertheless, there are no consistent performance gains from the uncertainty strategies.

### 6.2.2 Are Error-prone Examples Informative?

So far, we have established that the uncertainty strategies do tend to capture poor generation performance. However, in order for this to be reflected

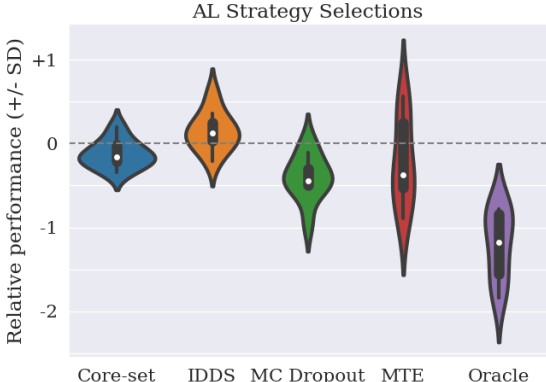

Figure 5: **Strategy selections by relative generation performance.** The plot compares AL strategies in terms of the relative model generation performance over the batches selected by the strategy. *Relative performance* is defined as the difference between the average performance over the batch and the average performance over the full unlabeled pool $U_D$, measured in standard deviations. Results are averaged across datasets. A large negative value indicates that the examples selected by the strategy are associated with poor model generation outputs.

in AL gains, the basic assumption that low performance examples are more useful to the model must be satisfied.

To test the validity of this assumption more directly, we implement an "illegal" AL strategy, named *Oracle*, that has direct access to the evaluation metric scores with respect to the ground-truth references, and selects the examples with the lowest evaluation scores (as seen in Fig. 5). If the aforementioned assumption is valid, *Oracle* is expected to yield optimal learning progress, as measured by the corresponding evaluation metric.

However, the results of this experiment, as shown in Appendix A.3, indicate that the *Oracle* strategy generally performs very poorly.

Thus, we see that the basic assumption of uncertainty sampling – that labeling examples with poor model output will be most beneficial for improving the model – does not necessarily hold in NLG.

To conclude, we have shown that uncertainty-based strategies are able to select error-prone instances; However, even optimal selection is not a guarantee of gains in model training, as demonstrated by the *Oracle* strategy.

## 7 Discussion

In this work, we examined the effectiveness of various active learning strategies across multiple NLG

tasks. Through rigorous experimentation and analysis, we have shown that no AL strategy systematically demonstrates a clear superiority over the random baseline in terms of NLG quality.

Our findings indicate that despite the potential promises and advancements in active learning techniques, when it comes to NLG, the inherent complexity and diversity of human language poses significant challenges. AL strategies, which aim to improve efficiency by actively selecting informative data points for labeling, may not effectively capture the intricacies of language structure, semantics, and context required for generating coherent and meaningful text.

Our results provide a wider and somewhat contrasting perspective to previous works. While previous papers had typically reported the effectiveness of AL on a specific dataset and task, our comprehensive analysis – spanning multiple datasets and tasks – suggests that the potential gains reported before are not very robust, hence do not show up in a consistent manner. Thus, while our experiments were done on a single base model and hyperparameter setting, they indicate that existing AL methods cannot be assumed to work out-of-the-box.

The failures of existing AL methods for NLG cannot easily be associated with a single underlying factor. More likely, there are multiple issues at play that violate some of the premises and assumptions of AL strategies that were designed with classification in mind. Some of these potential causes – for instance, the complex relation between model uncertainty and erroneous outputs – reflect a fundamental difference between the tasks of classification and generation. Others, however, may be more a question of degree. For instance, the output space for generation is overwhelmingly larger than that of a typical classification task, and is also characterized by a large degree of label imbalance, properties that may lead to difficulties in capturing informative examples. Notably, similar issues exist in some multi-label classification tasks, which also exhibit difficulties with leveraging AL (Wang and Liu, 2023; Wertz et al., 2022).

In this work we combine AL with a strong instruction-tuned model, highlighting the importance of the base model used. The behavior observed here, where much of the performance gain is achieved with a small number of training examples, is encouraging with respect to the practical scenario of a limited labeling budget; at the same

time, this may entail new AL approaches that focus on small batches or on bringing value to the selection of a single batch of few-shot instances.

In our view, the takeaway from our results is *not* that the paradigm of active learning should be abandoned. The goal of reducing manual annotation efforts remains as relevant as ever, all the more so for the particularly expensive annotation process associated with NLG. Rather, our hope is that these results will stimulate new efforts in devising novel AL strategies and approaches, ones that are specifically designed for the NLG scenario, and suited for strong instruction-tuned base models.

## Ethics and Broader Impact

This paper is submitted in the wake of a tragic terrorist attack perpetrated by Hamas, which has left our nation profoundly devastated. On October 7, 2023, thousands of Hamas terrorists infiltrated the Israeli border, launching a brutal assault on 22 Israeli villages. They methodically moved from home to home brutally torturing and murdering more than a thousand innocent lives, spanning from infants to the elderly. In addition to this horrifying loss of life, hundreds of civilians were abducted and taken to Gaza. The families of these abductees have been left in agonizing uncertainty, as no information, not even the status of their loved ones, has been disclosed by Hamas.

We fervently call for the immediate release of all those who have been taken hostage and urge the academic community to unite in condemnation of these unspeakable atrocities committed by Hamas. We call all to join us in advocating for the prompt and safe return of the abductees, as we stand together in the pursuit of justice and peace.

## Limitations

Generally, there are some inherent gaps between AL experiments such as those conducted here and the ultimate goal of achieving label efficiency with a human in the loop. As outlined in Margatina and Aletras (2023), prominent gaps between academic AL studies and practical AL applications include the potential effects of differing dataset qualities, as well as of temporal drifts in data distributions, that are characteristic of real-world data; additionally, while practitioners may pursue hyperparameter tuning for their trained model, this is not feasible in the context of a large AL study like the present work. Perhaps most crucially, given that

the AL experiments are performed on fully-labeled datasets, here we only look at the selection of examples for labeling, and not at the labeling process itself. Specifically, it is plausible that the examples selected by an AL strategy would prove to be more difficult for a human annotator, and/or that different labelers will write very different outputs for the selected instances. Such questions, which are highly relevant for NLG, are beyond the scope of this work.

In this study we report AL behavior for more practical labeling budgets, ranging between 20 and 1000 training examples. The effects of AL strategies when working with larger scales of labeled NLG data may be quite different than the pattern of results shown here.

Finally, as is common in NLG, we rely on automatic metrics to evaluate model performance. While these metrics are likely correlated to human judgements of task performance, the metrics may suffer from various artifacts and biases, and thus provide only a partial window into the true model performance.

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

# A  Appendix

## A.1  Datasets and Tasks

### Paraphrase Generation

Paraphrase generation datasets include pairs of an input text and its paraphrase, typically created by automatic alignment. To ensure high-quality candidate samples, we followed Dong et al. (2021), and kept only pairs where the BERTScore (Zhang et al., 2020) between the input and the paraphrase is higher than $0.8$.

*MSCOCO*: This dataset consists of 123K images, each being associated with at most five human-labeled captions (Lin et al., 2014). Following previous studies, we consider different captions of the same image as paraphrases. After applying the filtering we were left with more than 13K samples.

*Parabank1.0* and *Parabank2.0*: contain clusters of sentential paraphrases, produced from a bilingual corpus using lexical constraints to the NMT decoding procedure (Hu et al., 2019a) or negative constraints, inference sampling, and clustering (Hu et al., 2019b) respectively. These datasets are composed of an average of 5 paraphrases in every cluster and close to 80 and 100 million pairs in total. After filtering we were left with around 50K samples pairs in each of the datasets.

### Formality transfer

The Formality task is defined as the transition from informal to formal style.

*GYAFC*: The dataset was obtained from Rao and Tetreault (2018), and it contains 106K formal-informal pairs of sentences. Informal sentences were extracted from Yahoo Answers from two categories – "Entertainment & Music (E&M)" and "Family & Relationships (F&R)". The parallel formal sentences were produced with crowd workers. Due to its way of creation, it is considered a high-quality dataset, and hence no further filters were applied. Using the categories, we split GYAFC into two datasets, *GYAFC-E&M* and *GYAFC-F&R*, each with around 52K samples.

### Summarization

*DebateSUM*: This dataset (Roush and Balaji, 2020) consists of around 187K arguments, with corresponding evidence texts and extractive summaries that were compiled by experts (competitors within the National Speech and Debate Association). We consider the evidence extracts as the input texts and the arguments as the target abstractive summaries.

*Reddit TL;DR (openAI)*: The Reddit TL;DR dataset (Völske et al., 2017) contains more than 3 million reddit posts along with human-written summaries composed by the original posters. We use a subset of this dataset introduced by Stiennon et al. (2020), which consists of more than 123K posts and summaries with higher quality (removed duplications, removed summaries with certain levels of profanity, etc.). Summaries contain between 24 and 48 tokens.

### Question Generation

Question answering datasets are also used for the Question generation task, where given a context and an answer the model is asked to generate a question that leads to this answer.

*NewsQA*: a collection of more than 100K human-generated questions and answers. Questions are posed by crowd workers on a set of news articles from CNN, and the relevant span is annotated as the answer (Trischler et al., 2017).

*MLQA*: a multilingual question answering dataset, with questions generated by the crowd over English paragraphs from Wikipedia that were found to have corresponding parallel paragraphs in other languages. Professional translators then translate these questions into all target languages, and answer spans are annotated within the aligned contexts. In this work, we use the English subset only, which consists of 12K pairs (Lewis et al., 2020).

## A.2 Instructional Prompts

Table 2 reports all prompt templates used for the different tasks.

## A.3 Full Active Learning Plots

Figure 6 presents the active learning performance (§5) of all of the datasets tested.

Figure 7 depicts the full results for the *Oracle* strategy from the analysis section (§6.2.2).

## A.4 Statistical Significance Analysis

We perform a statistical significance analysis to evaluate the benefits of the different AL strategies in comparison to random selection.

Following Ein-Dor et al. (2020), we opt for the Wilcoxon signed-rank test due to its non-parametric nature. To calculate the p-value for a strategy $S$ over dataset $D$, we compare the performance of the relevant evaluation metric (Table 1) for all pairs $(S_{ij}, R_{ij})$, such that $R$ is the Random selection strategy, $i = (1...18)$ is the iteration index, and $j = (1...5)$ is the experiment repetition number. We apply a Bonferroni correction to adjust for the multiple strategies examined.

## A.5 Core-Set Adaptation

As stated in §4.4, we adapt the greedy Core-Set algorithm from Sener and Savarese (2018) for the scenario of starting with a zero-shot base model and an *empty* initial labeled pool $L_D$.

The original algorithm starts from a seed of labeled data, and then relies on it in order to greedily choose unlabeled examples one at a time to add to the labeled pool. In this work we begin with an empty pool $L_D$. Thus, for the first AL iteration of the Core-set strategy, we jump-start this process by randomly selecting a single unlabeled example to serve as the initial seed. We then use this single example as $L_D$ and apply the standard Core-Set algorithm for selecting $n_1 - 1$ instances, where $n_1$ is the batch size of the first iteration. The subsequent AL iterations of Core-Set are selected using the standard algorithm.

## A.6 Evaluation metrics

**Formality:** To obtain formality scores for model outputs, we train classifiers by fine-tuning DeBERTa-v3 once over the GYAFC-E&M dataset, and once over GYAFC-F&R (to accuracies of 92%); these classifiers are used to evaluate the level of formality of the unseen dataset, respectively. Then, we follow Xu et al. (2018) and use **G-Score** of the formality score and BERTScore (Zhang et al., 2020) with the reference text.

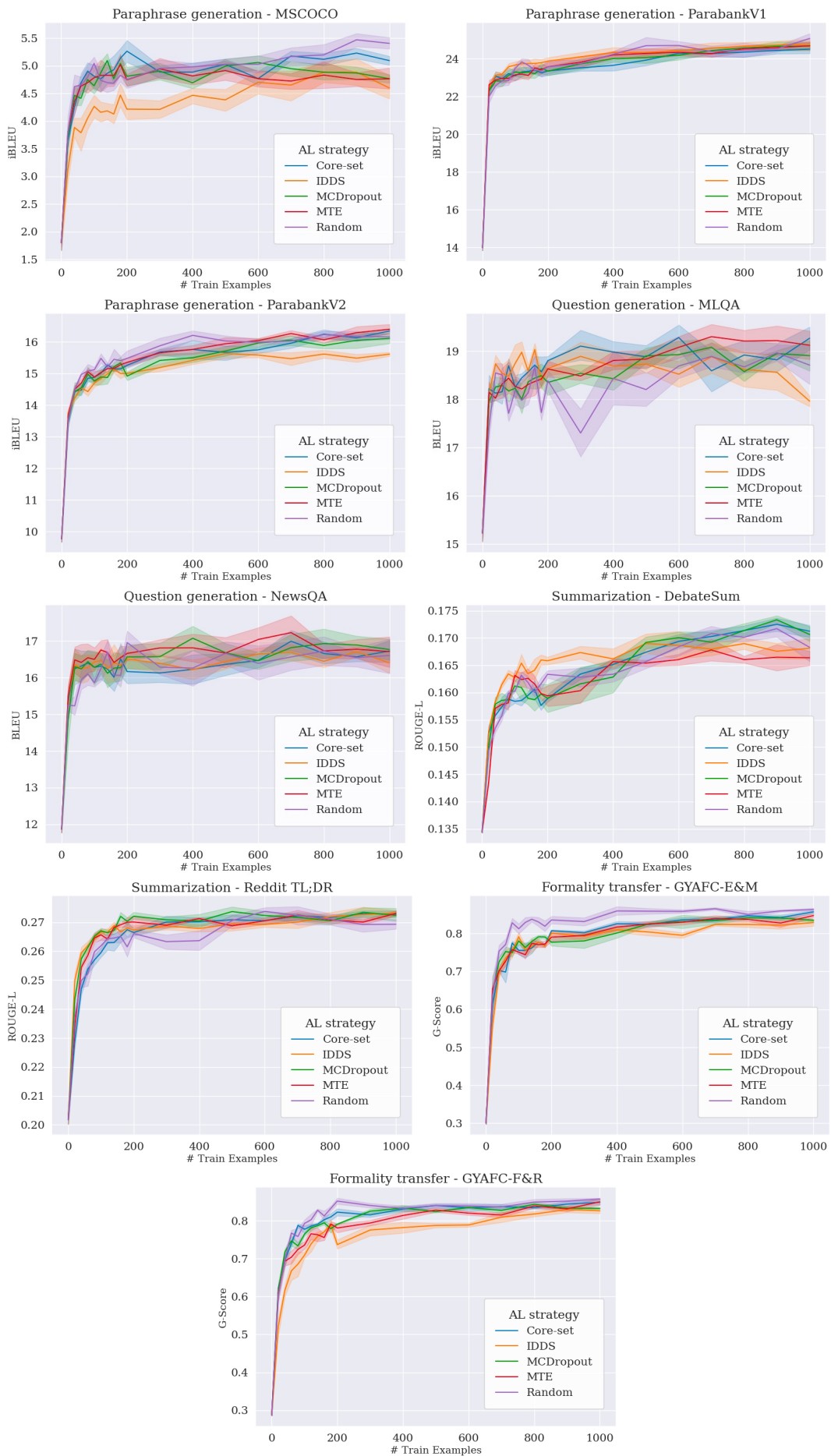

Figure 6: **AL performance.** The lines depict the evaluation performance of the different selection strategies along the AL process. Each line represents an average (± 95% Bootstrapped CI) over 5 experimental repetitions.

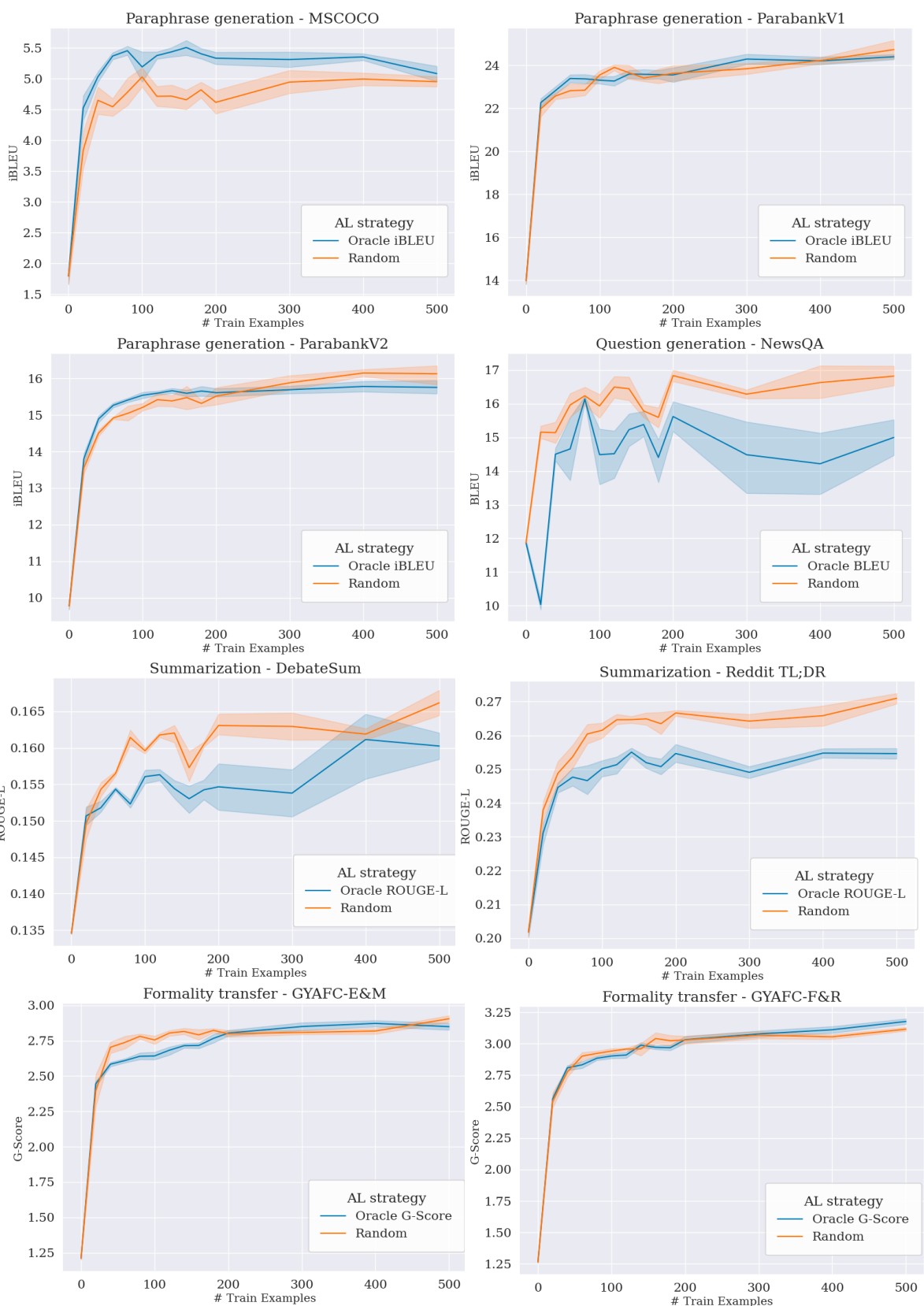

Figure 7: **AL performance of an "Oracle" strategy.** The lines depict evaluation performance along the AL process. The *Oracle* strategy is exposed to ground-truth labels, and selects examples based on poor performance on the automatic task evaluation metric. Each line represents an average (± 95% Bootstrapped CI) over 5 experimental repetitions. For the sake of presentation, in these plots we focus on the range of up to 500 labeled examples.

| Task | Prompt |
|------|--------|
| Paraphrase generation | Here is a text: {input text} |
| | Write a paraphrase for this text: |
| Summarization* | Here is a text: {document text} |
| | Write a short summary for this text: |
| Question generation | Here is some context: {context} |
| | And an answer: {answer} |
| | Given the context, write a question that leads to this answer: |
| Formality | Here is an informal text: {informal text} |
| | Write this text in a formal manner: |

Table 2: **Prompts**. *The summarization prompt was used only for the DebateSum dataset, as the Reddit TL;DR dataset includes its own prompts.