# OpenReview forum: "Active Learning for Natural Language Generation"
_EMNLP/2023/Conference — EMNLP 2023 Main_

### Official Review · Reviewer_74zp · 2023-07-30

**Soundness:** 4

**Excitement:**

4: Strong: This paper deepens the understanding of some phenomenon or lowers the barriers to an existing research direction.

**Paper Topic And Main Contributions:**

This paper analyses the challenge of applying Active Learning (AL) to Natural Language Generation (NLG). The AL technique selectively chooses the most informative examples to label, improving annotation efficiency. The paper presents a systematic study of AL for NLG tasks, considering a diverse set of tasks, multiple leading selection strategies, and a strong instruction-tuned model baseline. The study reveals that the performance of existing AL strategies is inconsistent, sometimes surpassing the baseline of random example selection. The authors' findings indicate that despite the potential of active learning techniques, human language's inherent complexity and diversity still pose a significant challenge to naively applying AL in NLG.

**Questions For The Authors:**

A: I am curious if you have an intuition as to why the Oracle strategy was worse than all the others? Is it because the outliers it selects are in fact mislabeled/corrupted examples?

**Reasons To Accept:**

- Systematic and Novel. The strengths of this paper lie in its systematic approach to studying AL for NLG, a highly relevant area of study. The authors conduct experiments across multiple tasks and datasets, testing and comparing known AL strategies. The authors are also the first to look into effective AL for NLG  (as far as I could find).

- Motivation for further research. The paper's findings provide valuable insights into the performance of existing AL strategies, highlighting their inconsistencies and the need for new methods. This could stimulate further research and innovation. The paper also offers a comparative analysis of the strategies, serving as a reference point for future studies. The authors' discussion on the inherent complexity and diversity of human language in the context of AL strategies could also benefit researchers working to gather data for LLMs.

**Reasons To Reject:**

- Limited model variation. The paper's findings are based on a single base model and a specific set of hyperparameters. This could limit the generalizability of the results, as the performance of AL strategies might vary with different models or hyperparameters.

- Gaps with real-world datasets. The authors acknowledge inherent gaps between AL experiments, such as those conducted in the paper and the goal of achieving label efficiency with a human in the loop. For instance, factors such as differing dataset qualities and temporal drifts in data distributions, characteristic of real-world data, are not accounted for in the experiments.


**Reproducibility:**

4: Could mostly reproduce the results, but there may be some variation because of sample variance or minor variations in their interpretation of the protocol or method.

**Reviewer Confidence:**

4: Quite sure. I tried to check the important points carefully. It's unlikely, though conceivable, that I missed something that should affect my ratings.

---

> ### Author Rebuttal · Authors · 2023-08-24
>
> Thank you for your helpful input.
> Indeed, as we note under Limitations, hyperparameter tuning and testing multiple models in the context of such a large-scale AL study is simply not feasible. The gaps between AL experiments and real world use, which we certainly acknowledge, are indeed a general issue with the field. We tried our best to account for what we can, but indeed these issues are beyond the scope of this study as well as of all classic AL empirical studies.
>
> A: Indeed, the key point here as we see it is the propensity to select outliers, which may be more extreme than those chosen by other strategies. We do not think this is necessarily one specific type of outliers - the Oracle can both tend to select mislabeled examples, examples with typos/tokenization issues etc., but also these can simply be examples that are uncharacteristic of the data distribution. In all of these different types of outliers we expect that training on these examples would not be very informative for the model, as they are not good representatives of the typical distribution of the data and task.

---

### Official Review · Reviewer_i9ma · 2023-08-04

**Typos Grammar Style And Presentation Improvements:** NA
**Soundness:** 4

**Excitement:**

3: Ambivalent: It has merits (e.g., it reports state-of-the-art results, the idea is nice), but there are key weaknesses (e.g., it describes incremental work), and it can significantly benefit from another round of revision. However, I won't object to accepting it if my co-reviewers champion it.

**Missing References:**

NA

**Paper Topic And Main Contributions:**

This paper is about Active Learning for Natural Language Generation. The main contributions of this paper are a systematic study of active learning for NLG, exploring a diverse set of tasks and selection strategies, an analysis of the selection behaviors of existing AL strategies, and highlighting some notable differences between the classification and generation scenarios.

**Questions For The Authors:**

NA

**Reasons To Accept:**

This paper presents a first systematic study of active learning for NLG, which is a novel contribution to the field. The authors explore a diverse set of tasks and selection strategies, and analyze the selection behaviors of existing AL strategies. They also highlight some notable differences between the classification and generation scenarios, which affect the performance of AL strategies. The results of their study indicate that the performance of existing AL strategies is inconsistent, surpassing the baseline of random example selection in some cases but not in others. Overall, this paper provides valuable insights into the application of active learning to NLG and lays the foundation for future research in this area.

**Reasons To Reject:**

While practitioners may pursue hyper-parameter tuning for their trained model, this is not feasible in the context of a large AL study like the present work. These limitations may affect the generalizability of the findings to real-world scenarios. However, these limitations are common to many academic studies and do not necessarily invalidate the contributions of the paper.

**Reproducibility:**

5: Could easily reproduce the results.

**Reviewer Confidence:**

3: Pretty sure, but there's a chance I missed something. Although I have a good feel for this area in general, I did not carefully check the paper's details, e.g., the math, experimental design, or novelty.

---

> ### Author Rebuttal · Authors · 2023-08-24
>
> Thank you for your helpful input.
> Indeed, as we note under Limitations, hyperparameter tuning in the context of a large-scale AL study is simply not feasible. But importantly, the capacity to tune hyperparameters is usually very limited in real-world settings as well, since where the goal is to minimize the amount of labeled data a validation set is not available (or is very small).

---

### Official Review · Reviewer_hX5h · 2023-08-11

**Paper Topic And Main Contributions:** 1. The paper presents a systematic st…
**Soundness:** 4

**Excitement:**

4: Strong: This paper deepens the understanding of some phenomenon or lowers the barriers to an existing research direction.

**Reasons To Accept:**

The paper describes multiple AL(Active Learning) strategies, divided into two categories: i.e. representativeness and informativeness. Also, uses these on 4 primary tasks: para-phrase generation, style transfer(formality), summarization, and question generation.
The paper uses an underlying pre-trained model i.e. Flan-T5 model hence avoids training data used in the model.
The paper presents AL performance using different strategies on Parabank1.0 and GYAFC-F&R dataset.

**Reasons To Reject:**

The paper poses multiple AL strategies to evaluate the performance of the model on unlabeled datasets.
1. After trying multiple techniques to label data, the author shows that the random strategy is the best strategy to use random data.
2. Author should devise novel algorithms for using unlabeled data to get better model performance after fine-tuning.

**Reproducibility:**

3: Could reproduce the results with some difficulty. The settings of parameters are underspecified or subjectively determined; the training/evaluation data are not widely available.

**Reviewer Confidence:**

4: Quite sure. I tried to check the important points carefully. It's unlikely, though conceivable, that I missed something that should affect my ratings.

---

> ### Author Rebuttal · Authors · 2023-08-24
>
> Thank you for your helpful input.
> Our goal was to understand the state of affairs in active learning for generation. As this is an empirical study, we did not know what results we would get, but indeed we were somewhat surprised that no strategy consistently outperformed the random strategy. As the focus of the paper was to evaluate and analyze the current state-of-the-art, devising new approaches is beyond the scope of this work. Given the results of our systematic study, the hope is that future work would offer novel algorithms for this setting. This is also something we are actively pursuing.

---

### Meta-Review · Area_Chair_pphn · 2023-09-19

**Recommendation:** 4

**Metareview:**

Summary:
The reviewers agree that the paper's positives including a systematic and novel study of active learning (AL) for natural language generation (NLG). Overall, the paper is said to lay the foundation for future NLG AL research and is a pioneering contribution to the field. It categorizes AL strategies into representativeness and informativeness and applies them to four primary tasks: paraphrase generation, style transfer (formality), summarization, and question generation. The study utilizes the Flan-T5 pre-trained model, avoiding the need for additional training data. The research provides valuable insights into the performance of existing AL strategies, highlighting their inconsistencies and the need for new methods. It motivates further research in NLG AL by offering a comparative analysis of strategies and addressing the complexities of human language.

The reviewers share the following limitations of the paper: (1) hyper-parameter tuning is not feasible in the context of a large active learning (AL) study like the one presented. This limitation may impact the applicability of the findings to real-world scenarios. (2) The study's findings are based on a single base model and specific hyperparameters and could restrict the generalizability of the results, as AL strategy performance may vary with different models or hyperparameter settings. (3) The authors recognize inherent gaps between AL experiments conducted in the paper and the ultimate goal of achieving label efficiency with human involvement in a real-world context. Factors such as varying dataset qualities and temporal shifts in data distributions, typical of real-world data, are not addressed in the experiments.

Reasons to Accept:
(1) The paper provides a systematic study of active learning (AL) for natural language generation (NLG), addressing a novel and relevant research area.
(2) It explores a diverse set of NLG tasks and selection strategies, shedding light on their performance and behaviors.
(3) The paper highlights the inconsistencies in the performance of existing AL strategies, providing valuable insights into the challenges and opportunities in NLG-related AL.
(4) The findings of this study can motivate further research and innovation in AL for NLG, serving as a reference point for future studies.

Reasons to Reject:
(1) The study primarily relies on a single base model and specific hyperparameters, potentially limiting the generalizability of the results to other models or settings.
(2) The experiments do not fully account for real-world dataset challenges, such as variations in dataset quality and temporal drifts in data distributions.

---

### Decision · Program_Chairs · 2023-10-07

**Decision:**

Accept-Main

**Comment:**

Summary:
The reviewers agree that the paper's positives including a systematic and novel study of active learning (AL) for natural language generation (NLG). Overall, the paper is said to lay the foundation for future NLG AL research and is a pioneering contribution to the field. It categorizes AL strategies into representativeness and informativeness and applies them to four primary tasks: paraphrase generation, style transfer (formality), summarization, and question generation. The study utilizes the Flan-T5 pre-trained model, avoiding the need for additional training data. The research provides valuable insights into the performance of existing AL strategies, highlighting their inconsistencies and the need for new methods. It motivates further research in NLG AL by offering a comparative analysis of strategies and addressing the complexities of human language.

The reviewers share the following limitations of the paper: (1) hyper-parameter tuning is not feasible in the context of a large active learning (AL) study like the one presented. This limitation may impact the applicability of the findings to real-world scenarios. (2) The study's findings are based on a single base model and specific hyperparameters and could restrict the generalizability of the results, as AL strategy performance may vary with different models or hyperparameter settings. (3) The authors recognize inherent gaps between AL experiments conducted in the paper and the ultimate goal of achieving label efficiency with human involvement in a real-world context. Factors such as varying dataset qualities and temporal shifts in data distributions, typical of real-world data, are not addressed in the experiments.

Reasons to Accept:
(1) The paper provides a systematic study of active learning (AL) for natural language generation (NLG), addressing a novel and relevant research area.
(2) It explores a diverse set of NLG tasks and selection strategies, shedding light on their performance and behaviors.
(3) The paper highlights the inconsistencies in the performance of existing AL strategies, providing valuable insights into the challenges and opportunities in NLG-related AL.
(4) The findings of this study can motivate further research and innovation in AL for NLG, serving as a reference point for future studies.

Reasons to Reject:
(1) The study primarily relies on a single base model and specific hyperparameters, potentially limiting the generalizability of the results to other models or settings.
(2) The experiments do not fully account for real-world dataset challenges, such as variations in dataset quality and temporal drifts in data distributions.